# Variable Expression of the Disialoganglioside GD2 in Breast Cancer Molecular Subtypes

**DOI:** 10.3390/cancers13215577

**Published:** 2021-11-08

**Authors:** Ramona Erber, Sareetha Kailayangiri, Hanna Huebner, Matthias Ruebner, Arndt Hartmann, Lothar Häberle, Julia Meyer, Simon Völkl, Andreas Mackensen, Laura Landgraf, Carol I. Geppert, Rüdiger Schulz-Wendtland, Matthias W. Beckmann, Peter A. Fasching, Nicole Farwick, Claudia Rossig, Paul Gass

**Affiliations:** 1Institute of Pathology, Erlangen University Hospital, Friedrich Alexander University of Erlangen–Nuremberg (FAU), Comprehensive Cancer Center Erlangen-EMN, 91054 Erlangen, Germany; arndt.hartmann@uk-erlangen.de (A.H.); laura.landgraf@uk-erlangen.de (L.L.); carol.geppert@uk-erlangen.de (C.I.G.); 2Department of Pediatric Hematology and Oncology, Münster University Hospital, 48149 Münster, Germany; Sareetha.Kailayangiri@ukmuenster.de (S.K.); Nicole.Farwick@ukmuenster.de (N.F.); rossig@ukmuenster.de (C.R.); 3Department of Gynecology and Obstetrics, Erlangen University Hospital, Friedrich Alexander University of Erlangen–Nuremberg (FAU), Comprehensive Cancer Center Erlangen-EMN, 91054 Erlangen, Germany; Hanna.Huebner@uk-erlangen.de (H.H.); Matthias.Ruebner@uk-erlangen.de (M.R.); Lothar.Haeberle@uk-erlangen.de (L.H.); Julia.Meyer@uk-erlangen.de (J.M.); Matthias.Beckmann@uk-erlangen.de (M.W.B.); Peter.Fasching@uk-erlangen.de (P.A.F.); 4Biostatistics Unit, Department of Gynecology and Obstetrics, Erlangen University Hospital, Friedrich Alexander University of Erlangen–Nuremberg (FAU), 91054 Erlangen, Germany; 5Department of Internal Medicine 5, Hematology/Oncology, Erlangen University Hospital, Friedrich Alexander University of Erlangen–Nuremberg (FAU), Comprehensive Cancer Center Erlangen-EMN, 91054 Erlangen, Germany; Simon.Voelkl@uk-erlangen.de (S.V.); Andreas.Mackensen@uk-erlangen.de (A.M.); 6Institute of Diagnostic Radiology, Erlangen University Hospital, Friedrich Alexander University of Erlangen–Nuremberg (FAU), Comprehensive Cancer Center Erlangen-EMN, 91054 Erlangen, Germany; Ruediger.Schulz-Wendtland@uk-erlangen.de; 7Division of Hematology/Oncology, Department of Medicine, David Geffen School of Medicine, University of California, Los Angeles, CA 90095, USA

**Keywords:** GD2, breast cancer, immunofluorescence, immunohistochemistry, prognosis, disialoganglioside

## Abstract

**Simple Summary:**

GD2 is an antigen that is tumor-specific and can be used as a target for specific immunotherapies. Since the knowledge about GD2 in breast cancer is limited, we analyzed the frequency of GD2 expression in breast cancer using two different staining methods and the impact of GD2 expression on the survival of breast cancer patients. GD2 expression was found in more than 50% of breast cancer cases, with the highest frequency in hormone receptor-positive tumors. GD2 expression was not significantly associated with patient outcome. Unlike previous studies with smaller sample sizes that lacked correlation with clinical data, this study includes a larger cohort and associations with survival data and shows that GD2 is expressed on human breast cancer cells, providing a potential target for immunotherapies (e.g., anti-GD2 antibodies or GD2 CAR T cells), that are currently undergoing clinical testing.

**Abstract:**

The disialoganglioside GD2 is a tumor-associated antigen that may allow for the application of targeted immunotherapies (anti-GD2 antibodies, GD2 CAR T cells) in patients with neuroblastoma and other solid tumors. We retrospectively investigated GD2 expression in a breast cancer cohort, using immunohistochemistry (IHC) and immunofluorescence (IF) on tissue microarrays (TMAs), and its impact on survival. GD2 expression on IHC (*n* = 568) and IF (*n* = 503) was investigated in relation to subtypes and patient outcome. Overall, 50.2% of the 568 IHC-assessed samples and 69.8% of the 503 IF-assessed samples were GD2-positive. The highest proportion of GD2-positive tumors was observed in luminal tumors. Significantly fewer GD2-positive cases were detected in triple-negative breast cancer (TNBC) compared with other subtypes. The proportion of GD2-expressing tumors were significantly lower in HER2-positive breast cancer in comparison with luminal tumors on IF staining (but not IHC). GD2 expression of IHC or IF was not significantly associated with disease-free or overall survival, in either the overall cohort or in individual subtypes. However, GD2 expression can be seen in more than 50% of breast cancer cases, with the highest frequency in hormone receptor-positive tumors. With this high expression frequency, patients with GD2-positive advanced breast cancer of all subtypes may benefit from GD2-targeting immunotherapies, which are currently subject to clinical testing.

## 1. Introduction

Tremendous progress has been made in the field of breast cancer therapy over the last few decades [1,2,3,4,5]. Immunotherapies have recently been shown to improve the outcome for patients with various cancers. In addition to checkpoint inhibitors, immune therapies include chimeric antigen receptor gene-modified T cells (CAR T cells) and antibody therapies against antigens selectively expressed on the cell surface of tumor cells [6,7,8]. In 2009, the National Cancer Institute listed the disialoganglioside GD2 (in Svennerholm’s nomenclature system [9]) as the 12th most promising tumor antigen [10]. It belongs to a large family of more than 180 known ganglioside biomolecules that are composed of glycosphingolipids (i.e., glycosylated lipid molecules) that contain a variety of carbohydrates and sialic acid components [9,11]. The expression of individual gangliosides is tissue-specific and varies for different developmental stages. GD2 is mostly, but not exclusively, located in the outer leaflets of plasma membranes [11] and is widely expressed during embryonal development. After birth, GD2 expression is mostly restricted to the neural tissue of the central nervous system and peripheral nerves, as well as to skin melanocytes [12,13]. GD2 can also be frequently found in some solid tumors in both pediatric and adult patients, especially in neuroblastoma [14] and melanoma [15], as well as some sarcomas, e.g., Ewing’s sarcoma [16], osteosarcoma [17,18], and uterine leiomyosarcoma [19]. Due to its selective overexpression in tumor cells, GD2 has attracted interest as a potential immunotherapeutic target. Several therapeutic anti-GD2 antibodies have been investigated in clinical studies [20,21,22], leading to the approval of the chimeric anti-GD2 antibodies dinutuximab and dinutuximab beta for the treatment of neuroblastoma in combination with interleukin-2. In high-risk neuroblastoma, GD2-specific antibodies, that are used to maintain remission after induction therapy and stem cell transplantation, have resulted in a superior 2-year event-free survival and 2-year overall survival in comparison with standard maintenance therapy [20]. To further increase the survival rates through the use of more potent effector mechanisms, GD2-targeting CAR T cells have been developed and are being tested in clinical trials [23].

To date, there has been limited research on GD2 expression in breast cancer and its impact on the prognosis. Although GD2 is weakly expressed or absent in normal breast tissue [24], it has been suggested that it may represent a marker of breast cancer stem cells [25], and it has been found to be associated with more aggressive breast cancer subtypes and with breast cancer-initiating cells [26,27,28]. GD2 targeting that uses antibodies or CAR T cells might therefore be a future option in immunotherapy for advanced breast cancer [29].

Previous studies that have analyzed GD2 in primary breast cancer have been limited by small sample sizes and their lack of correlation with the clinical data. The present study investigated the frequency of GD2 expression in intrinsic breast cancer subtypes and its impact on the prognosis.

## 2. Materials and Methods

### 2.1. Patient Selection

The Bavarian Breast Cancer Cases and Controls (BBCC) study is a case-control study investigating molecular and epidemiological breast cancer risk and prognostic and predictive parameters, that is being conducted at the University Breast Center for Franconia (Bavaria, Germany) [30,31,32]. Patients were eligible for inclusion if they satisfied the following criteria: if they were at least 18 years of age and had received a diagnosis of invasive breast cancer. Formalin-fixed, paraffin-embedded (FFPE) tissue from the primary tumors was available for 894 of these patients with an initial diagnosis from 1997 to 2007 in order to construct a tissue microarray (TMA). For further analysis, specific patient groups were excluded: male patients, female patients with bilateral breast cancer at the initial diagnosis, patients with metastases present at the initial diagnosis or with insufficient survival times (disease-free survival/overall survival less than 1 day; *n* = 105), and patients without an assessable GD2 status (Figure 1). The final sample size consequently comprised 568 patients for an analysis of GD2 using immunohistochemistry (IHC) and 503 patients for the GD2 immunofluorescence (IF) analysis (Figure 1). The ethics committee of the Medical Faculty of Erlangen University Hospital approved this retrospective study (ref. numbers 2700 and 297_17 Bc).

### 2.2. Clinical Data and Histopathological Assessment

The process of data collection has been described in detail elsewhere [33,34]. Briefly, all clinical and histopathological data were documented prospectively in an annually audited, certified database [35,36]. Data on histological tumor type, tumor grading, estrogen receptor status, progesterone receptor status, and HER2 status were obtained from the original routine pathology reports. Detailed grading/IHC protocols and definitions of the subtypes are listed in the Appendix A.

### 2.3. Assessment of GD2 Expression

To identify the optimal method of assessing GD2 expression in breast cancer, both IHC and IF detection methods were used on 2-µm thick sections of TMAs containing invasive breast cancer or a non-neoplastic breast epithelium (for construction of the TMAs, see the Appendix A), stained with the same anti-human disialoganglioside GD2 monoclonal mouse antibody (clone 14.G2a; 554272, BD Pharmingen^TM^, Heidelberg, Germany) [19].

### 2.4. GD2 Staining by Immunohistochemistry

IHC was performed manually in accordance with the institute’s standards and the manufacturer’s instructions. After heat-induced epitope retrieval (HIER) using a Tris/EDTA buffer, pH 9 (Agilent/Dako, Santa Clara, CA, USA) at 120 °C for 1 min, antibody incubation with the primary GD2 antibody (dilution 1:100) was performed at room temperature overnight. The process of the antibody binding to the antigen was visualized using an avidin-biotin complex-based peroxidase system (Vectastain^®^ Elite^®^ ABC HRP Kit (peroxidase, mouse IgG), Vector Laboratories, Burlingame, CA, USA) and subsequent counterstaining with hematoxylin. The FFPE tissue from a neuroblastoma and from an invasive GD2-positive breast cancer sample were used as positive controls. As a negative control, a buffer was applied instead of the antibody. GD2 expression was scored semi-quantitatively for each breast cancer TMA core by a pathologist who was blinded to any patient information. The intensity was classified in a four-tiered fashion into no staining at all (0), weak (1+), moderate (2+), or strong (3+) staining. The percentage of GD2-positive tumor cells was assessed as a continuous parameter (0–100%). TMAs containing a non-neoplastic breast epithelium were counted in the same manner.

### 2.5. GD2 Staining by Immunofluorescence

The GD2 monoclonal mouse antibody clone 14.G2a was also used in an IF protocol [16], modified for FFPE tissue. After the deparaffinization of 2-µm thick sections of each breast cancer (BBCC) TMA with xylene and ethanol in decreasing concentrations, HIER was performed with an ethylenediamine tetraacetic acid (EDTA) buffer (pH 8.5) in a gas stove. Protein blocking was performed using goat serum (1 h at room temperature). The primary antibody (GD2) was incubated at a dilution of 1:25 at 37 °C for 1 h, with an IgG2a antibody (dilution 1:50) serving as the negative control. After the incubation of the secondary antibody (Opal Polymer HRP Ms + Rb; Perkin-Elmer, Waltham, MA, USA) for 10 min at room temperature, Opal 520 (dilution 1:200) and 4′,6-diamidino-2-phenylindole (DAPI) were applied. Washing was conducted between the steps using tris-buffered saline with Tween 20 (TBS-T). Sections of GD2-positive Ewing sarcomas and breast cancers were used as positive controls. The GD2 expression was scored semi-quantitatively using the same method as used for the GD2 IHC stains.

### 2.6. GD2 Flow Cytometry

Flow cytometry was also performed to investigate possible membranous expression of GD2. The protocol, including the gating strategy (Appendix A) is detailed in the Appendix A.

### 2.7. Statistical Analysis

The primary objective of this analysis was to investigate whether there is a significant association between breast cancer subtypes and GD2 expression. For this purpose, Kruskal-Wallis tests and, in case of significance, post-hoc Wilcoxon rank-sum tests were calculated.

The secondary objective was to investigate whether the biomarker GD2 had a prognostic impact on disease-free survival (DFS) or overall survival (OS) in addition to well-known prognostic patient and tumor characteristics. DFS was defined as the time from the date of primary diagnosis to the earliest date of disease progression (distant metastasis, local recurrence, or death of any cause) or the date of censoring. Patients who were lost to follow-up before the maximum observation time of 10 years, or who were disease-free after the maximum observation time, were censored at the last date on which they were known to be disease-free or at the maximum observation time. The OS was defined in a similar way.

A multivariate Cox regression model (the basic model) was fitted with the following predictors: age at diagnosis (continuous), body mass index (continuous), the tumor stage (ordinal: T1 to T4), lymph-node status (categorical: N0, N1) and subtype (categorical: TNBC, luminal A, luminal B, HER2+). The proportional hazards assumptions were checked for both outcomes using the Grambsch and Therneau method [37]. Where the proportional hazards assumption was violated, a stratification for the corresponding predictors was implemented in the models.

Subsequently, an additional Cox regression model (the full model) was fitted, containing the predictors from the basic model and the biomarker GD2, as well as the interaction between GD2 and subtypes. The full model was compared to the basic model using a likelihood ratio test (LRT). If the test or the interaction term was not found to be significant, the interaction term was excluded from the full model in order to calculate an adjusted hazard ratio (HR) for GD2. Patients with missing survival information and missing values for the biomarker of interest (GD2) were excluded from the analysis. Missing values for other predictors were imputed, as done previously [38].

For the sensitivity analyses, unadjusted HRs were estimated using univariate Cox regression models for the GD2 expression (0 vs. >0) and percentage. Survival rates for the binary intensity GD2 variable were estimated using the Kaplan–Meier product limit method. For the subtypes (i.e., TNBC, luminal A, luminal B, HER2+), the Kaplan-Meier curves for GD2 (IHC) were produced for illustrative purposes.

All of the analyses were carried out separately for GD2 expression, determined using either the IHC or IF method. In addition, GD2 was included as continuous (percentage) and binary intensity (0 vs. >0) variables in all of the analyses. To assess the agreement of the two continuous percentage variables, Lin’s concordance correlation coefficient and the corresponding 95% confidence interval (CI) were calculated [39]. All of the tests were two-sided, and a *p* < 0.05 was regarded as statistically significant. Calculations were carried out using the R system for statistical computing (version 3.6.1; R Development Core Team, Vienna, Austria, 2019).

## 3. Results

### 3.1. Patients

After the exclusion of patients on the basis of the criteria mentioned above, the final sample cohort comprised of 789 patients, 568 of whom had samples analyzed for GD2 expression using IHC and 503 of whom had samples analyzed for GD2 expression using IF (Figure 1). The baseline characteristics of the study cohort are listed in Table 1.

### 3.2. GD2 Expression in Invasive Breast Cancer

GD2 was found to be expressed in 285 of the 568 (50.2%) breast cancer samples using IHC and in 351 of the 503 (69.8%) breast cancer samples using IF. GD2 IHC showed a cytoplasmic, mostly paranuclear granular pattern. A distinct membranous staining, clearly differing from the cytoplasmic expression, was not distinguishable on IHC (Figure 2a,b). The same was observed for the GD2 IF (Figure 2c,d). An evaluation of the GD2 expression was therefore performed for each tumor cell as a whole (not subdivided into different subcellular locations). To confirm membrane staining, which is a prerequisite for antibody-based targeted therapeutics, GD2 expression was assessed on freshly isolated single-cell suspensions obtained from four breast cancer samples that had previously tested positive for GD2 expression on IHC/IF. GD2 was also expressed on the surface of the tumor cells in all four samples (Appendix A). Moreover, GD2 expressions in breast cancer cell lines analyzed by both flow cytometry and GD2 immunohistochemistry of corresponding cell line FFPE blocks were compared (Appendix A).

The distribution of GD2 intensities differed significantly among the breast cancer subtypes with both staining methods (each *p* < 0.0001; Table 2, Appendix A). On IHC, 18.5% of the TNBC samples, 59.7% of luminal A, 53.0% of luminal B, and 45.5% of the HER2+ BC samples were GD2-positive at low to high intensities. The difference between the TNBC subgroup and the other three groups was statistically significant, while the others did not differ significantly among each other. The IF method detected GD2 expression in 44.9% of TNBC, 78.1% of luminal A, 74.7% of luminal B, and 55.4% of HER2+ subtypes. The TNBC group differed significantly from the luminal A and B subtypes, while the HER2+ group varied significantly, but not drastically compared to the other groups.

The distribution of these percentages over the breast cancer subtypes can be seen in the box plots in Appendix A. For both of the staining methods, the percentage of GD2-positive tumor cells differed significantly across the four intrinsic subgroups (each *p* < 0.0001). For GD2 IHC, the distribution of the TNBC group significantly differed from all three of the other groups. For GD2 IF, the TNBC and HER2+ subgroups differed significantly from the luminal A and B categories.

Using Lin’s concordance correlation coefficient, the agreement of GD2 percentage values measured using the IHC and IF methods was found to be 0.655 (95% CI, 0.60 to 0.70).

### 3.3. Disease-Free and Overall Survival of Patients with GD2-Positive Versus GD2-Negative Invasive Breast Cancers

A Kaplan–Meier survival analysis revealed that GD2 expression on tumor cells does not have a prognostic impact on DFS in breast cancer patients overall (Figure 3; LRT: *p* = 0.49 for IHC binary, *p* = 0.92 for IHC percentage, *p* = 0.66 for IF binary, *p* = 0.39 for IF percentage). Similarly, GD2 did not show any significant prognostic impact in relation to OS (LRT: *p* = 0.37 for IHC binary, *p* = 0.43 for IHC percentage, *p* = 0.90 for IF binary, *p* = 0.53 for IF percentage; Figure 3, Table 3). A comparative analysis of the survival between GD2-positive and GD2-negative tumors among patient subgroups defined by breast cancer subtypes was limited by the small number of cases in the TNBC and HER2+ subsets and did not reveal any associations between GD2 positivity and DFS (Appendix A) or OS (Appendix A).

### 3.4. GD2 Expression in Non-Neoplastic Breast Parenchyma

To not discount the co-expression of GD2 on adjacent normal tissue, 130 TMA cores with sufficient normal surrounding breast parenchyma were analyzed. The normal tissue was GD2-negative in 99 samples (76.2%). The remainder (31/130, 23.8%) showed a rare focal GD2 positivity in the breast epithelium, ranging from a weak to strong expression and up to 5% of non-neoplastic epithelial cells (Appendix A). Within these positive cases, 17/31 (54.8%) and 24/31 of the samples (77.4%) showed an expression of GD2 in only 1% and 1–2% of all the assessable epithelial cells, respectively. In addition, endothelium as well as fatty and connective tissue did not harbor any GD2 positivity.

## 4. Discussion

This retrospectively conducted breast cancer study, based on FFPE tissue of invasive breast cancer samples, investigated the expression of the ganglioside GD2 and its impact on survival. Overall, 50.2% and 69.8% of the assessable breast cancer samples in the cohort were found to be GD2-positive using IHC or IF, a finding that is in line with another study that reported GD2 positivity in 59% of breast cancer cases using IHC [27]. Both staining methods indicated both a membranous expression (which was confirmed in single cases by flow cytometry) and cytoplasmic staining, and particularly a perinuclear granular “Golgi-like” pattern. This partly cytoplasmic expression pattern can be explained by the fact that the synthesis and the modification of gangliosides occurs in the endoplasmic reticulum and Golgi apparatus [11,40]. In line with the present findings, Orsi et al. reported on cytoplasmic staining combined with membranous staining but did not distinguish these compartments when evaluating the GD2 IHC [27].

Among the subgroups, GD2 expression was observed predominantly in luminal breast cancer. Independently of the staining methods, the lowest proportion of GD2-expressing samples was found in TNBC compared to the other breast cancer subtypes. This is in contrast to the findings of another study, which reported an association of GD2 with TNBC and age >78 years in a univariate analysis. However, the cohort included was relatively small (*n* = 63) and the majority of the samples showed only weak staining. In addition, an association with TNBC was not confirmed in the multivariate statistical model [27]. A different antibody was also used. In the present study, we decided to use the clone 14.G2a, which has been described and established in several studies [16,19,41,42] and is also used for therapeutic purposes [20,21,43]. Another difference that may explain the divergent results is that the proportion of metaplastic carcinomas was small in the present patient group (*n* = 5). In contrast, 22% of the cohort in the study by Orsi et al. had metaplastic carcinomas, with 79% expressing GD2 [27]. These tumors show distinct biological behavior and have an extremely poor prognosis. Whether GD2-specific therapy might be an option for this specific subset has yet to be clarified.

The comparison of the two GD2 staining methods, IHC and IF, showed a good, but not an excellent, correlation level of 0.655. A higher rate of GD2-positive tumors was detected with IF. However, IHC has the advantage of being easier to implement in the routine diagnosis of pathology laboratories, whereas IF requires special reagents, protocols, and microscopes. The disadvantages of IF include a higher rate of tissue cores that were displaced from the slides during the staining process and samples with strong auto-fluorescence, resulting in cores not being assessable. In addition, weak auto-fluorescence may hamper an accurate evaluation when using fluorescence microscopes that do not automatically fade these artefacts out.

The expression of GD2 was correlated with the outcome for the patients, with special attention being provided to the different intrinsic breast cancer subtypes. There were no significant associations between the GD2 expression and outcome for patients in the overall cohort (DFS and OS), nor was there a statistical impact of the GD2 expression on DFS or OS in the intrinsic breast cancer subtypes. When the four individual staining intensities were analyzed separately, we found a slight trend toward 3+ GD2-positive tumors, associated with poorer DFS and OS, particularly in the luminal A subtype. However, this observation requires validation in another study cohort with larger numbers. With regard to the outcome analysis, it should be noted that the sample sizes for each of the subtypes, particularly the TNBC and HER2+ categories, were too small for definitive conclusions to be drawn.

The present results might argue against a prognostic significance of GD2 expression in breast cancer. However, this issue has to be analyzed in further breast cancer cohorts. Nevertheless, GD2 in breast cancer might be a suitable tumor-associated antigen that could be targeted on the cell surface in patients with a poor prognosis according to established clinical and pathological risk factors. The advantages of GD2 in comparison with other tumor-associated antigens include the expression levels of GD2 on the cell surface being either low or undetectable in most organs/tissues after birth and that GD2 covers only 1–2% of the surface gangliosides on non-neoplastic cells [13]. However, GD2 antibody therapy can cause substantial toxicities in children, including capillary leak syndrome and neuropathic pain [20,21]. With increasing clinical experience, GD2 antibody-related toxicities have become manageable, and GD2 antibody infusions have become the current standard maintenance therapy for preventing relapse in patients with high-risk neuroblastoma [20,21]. The use of GD2-specific CAR T cells in initial clinical studies has not yet been associated with severe toxicities [43,44], but the efficacy of the treatment has, to date, been limited and the strategy clearly needs further refinement. In addition, carbohydrate mimetic peptide vaccines and theranostic pre-targeted radioimmunotherapy of the GD2 antigen on the cell surface may be promising future therapies [45,46,47].

To address the risks of on-target/off-tumor toxicities of GD2 targeting, GD2 expression in non-neoplastic breast parenchyma was additionally investigated. Apart from vessels, fatty tissue, and connective tissue, which were all negative for GD2, there were some cases (23.8%) that displayed minimal expression in non-atypical breast epithelium cells. In these samples, however, there were only single cells with Golgi-like, but not membranous GD2 expression. Adverse side effects in non-neoplastic breast parenchyma are therefore unlikely. On the basis of the observation that GD2 expression in breast cancer cells is associated with stem cell-like behavior and with activated NF-κB signalling [48], a small-molecule inhibitor of NF-κB has been investigated as a therapeutic agent in preclinical studies. The inhibitor did in fact reduce GD2 synthesis as well as tumor cell growth and the migration of breast cancer cells in vitro and presented antitumor activity in vivo. The molecular targeting of pathways associated with GD2 synthesis and expression would be most effective if GD2 was indeed a marker of aggressive and/or tumor-initiating cells, with critical contributions to tumor growth and metastasis. The finding that GD2 expression is not associated with the outcome in the present cohort of patients argues against the relevance of the ganglioside in the biological behavior of breast cancer cells, but the effect might have been concealed by the efficacy of treatments that the patients had received. Indeed, detailed experimental studies are required in order to address the biological role of GD2 in breast cancer.

One disadvantage of the present study is the fact that the survival analysis in the different intrinsic subgroups was limited, due to the small number of cases per subtype. Moreover, more than 200 cases had to be excluded from each IHC and IF analysis due to the technical issues of TMA evaluation (e.g., inadequate tumor tissue recognizable, tumor core washed off, autofluorescence). Hence, the existence of a selection bias cannot be excluded completely. Another issue that should be mentioned is that it was not possible to make any predictions about the pathological complete response, since most patients in the cohort were treated before the advent of neoadjuvant therapy. As shown above, IF and IHC did not provide comparable results of GD2 expression. We used an antibody that has been developed for flow cytometry, IF, and IHC on frozen tissue but not on FFPE tissue. This may explain the differences in expression levels. Hence, further studies that address the optimal evaluation of GD2 detection in breast cancer are needed. Furthermore, we experienced difficulties in separating a distinct membranous staining—as prerequisite for antibody-based targeted therapeutics—from the predominant cytoplasmic GD2 expression. Although we confirmed the occurrence of membranous GD2 staining in four immunohistochemically GD2-positive breast cancer samples, we can suppose, but not guarantee, the same for each tumor of the investigated retrospective cohort due to a lack of fresh tissue and, the impossibility of using flow cytometry for confirmation. We detected a cytoplasmic, mostly paranuclear granular pattern which was partly reminiscent of a “Golgi-like” pattern. However, the literature on subcellular GD2 localization in breast cancer cells is limited. Orsi et al. described cytoplasmatic along with membrane staining by IHC [27], but did not provide further details. Therefore, future studies should aim to investigate the precise subcellular GD2 localization for a more sophisticated staining interpretation. GD2 expression did not achieve prognostic significance with regard to patients’ outcome. Using a multi-variable score combining the intensity and percentage of GD2-positive tumor cells (e.g., Allred-score) with external validation might improve the prediction of prognosis.

## 5. Conclusions

In conclusion, the present study revealed the presence of GD2 expression in a high proportion of breast cancer samples, with a significantly higher proportion of GD2-positive tumors in luminal versus triple-negative breast cancer. Although further studies are required to confirm these findings, it can be understood that only a few of the TNBC patients are likely to be suitable for future GD2-specific immunotherapy. Whether or not GD2-based therapies present an option for patients with luminal A or B subtypes and HER2+ cancers, depends on results obtained and analyzed in future trials.

## Figures and Tables

**Figure 1 cancers-13-05577-f001:**
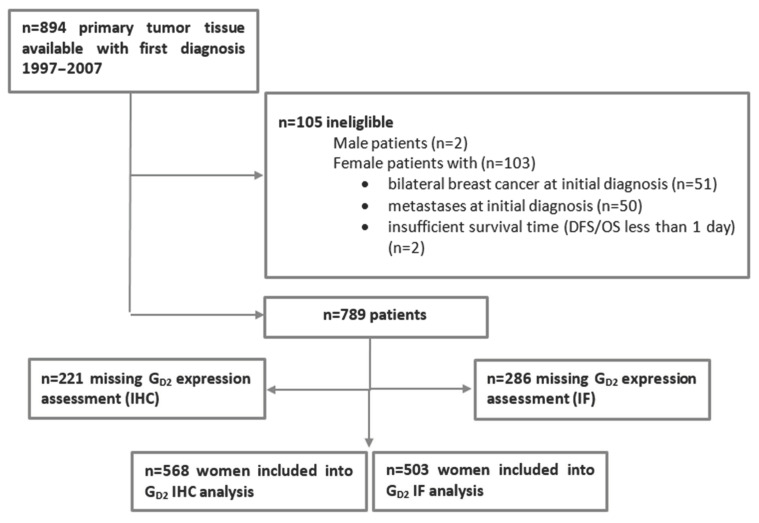
Patient selection and exclusion criteria. DFS, disease free-survival; IF, immunofluorescence; IHC, immunohistochemistry; OS, overall survival.

**Figure 2 cancers-13-05577-f002:**
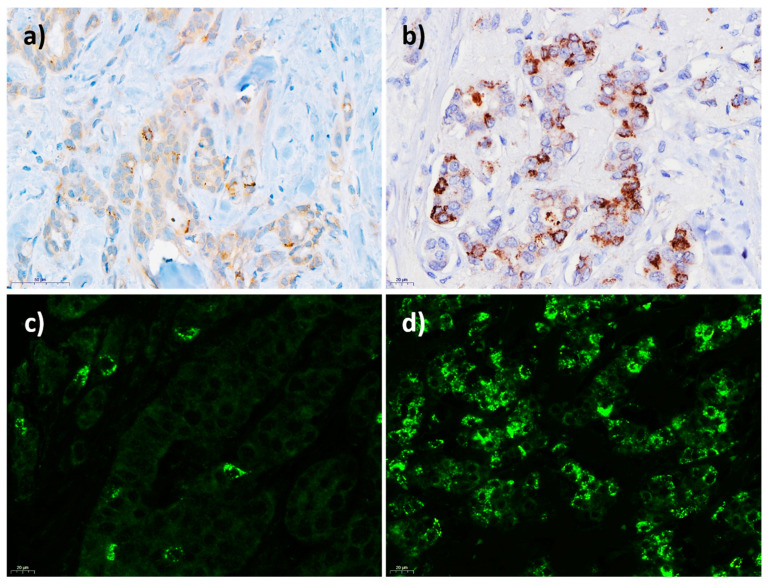
(**a**,**b**) GD2 immunohistochemistry (IHC) in breast cancer: (**a**) weak to intermediate GD2 positivity in few breast cancer cells; (**b**) intermediate to strong GD2 positivity. (**c**,**d**) GD2 immunofluorescence (IF) in breast cancer: (**c**) intermediate to strong GD2 positivity in a few breast cancer cells; (**d**) GD2 positivity in more than 50%.

**Figure 3 cancers-13-05577-f003:**
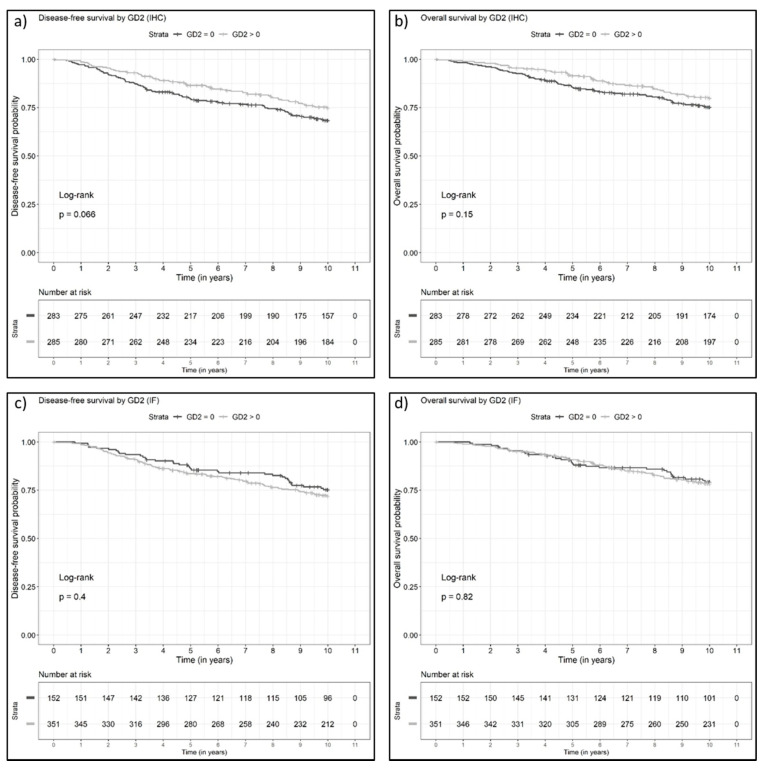
(**a**,**b**) Kaplan–Meier curves for (**a**) disease-free survival and (**b**) overall survival (OS) in patients with negative (0) and positive (>0) GD2 immunohistochemistry expression. (**c**,**d**) Kaplan–Meier curves for (**c**) disease-free survival and (**d**) overall survival in patients with negative (0) and positive (>0) GD2 immunofluorescence expression. IF, immunofluorescence; IHC, immunohistochemistry.

**Table 1 cancers-13-05577-t001:** Baseline characteristics of the study cohort with eligible GD2 values (based on immunohistochemistry and immunofluorescence) and results of GD2 assessment.

Characteristic	IHC (*n* = 568)	IF (*n* = 503)
Age at diagnosis (years; mean, SD)	58.2 (12.4)	58.2 (12.5)
BMI (kg/m^2^; median, IQR)	25.4 (22.7–28.6)	25.4 (22.7–28.7)
Grading		
G1	70 (12.3)	61 (12.1)
G2	367 (64.6)	320 (63.6)
G3	131 (23.1)	122 (24.3)
Lymph-node status		
N0	340 (59.9)	304 (60.4)
N1	228 (40.1)	199 (39.6)
Tumor stage		
T1	297 (52.3)	273 (54.3)
T2	220 (38.7)	186 (37.0)
T3	28 (4.9)	24 (4.8)
T4	23 (4.0)	20 (4.0)
Breast cancer subtype		
TNBC	76 (13.4)	69 (13.7)
Luminal A	228 (40.1)	196 (39.0)
Luminal B	198 (34.9)	182 (36.2)
HER2	66 (11.6)	56 (11.1)
Histologic subtype		
NST/IDC	417 (73.4)	366 (72.8)
ILC	70 (12.3)	66 (13.1)
Medullary pattern	21 (3.7)	23 (4.6)
Tubular	15 (2.6)	14 (2.8)
Other/Unknown	14 (2.5)	23 (4.6)
Micropapillary	13 (2.3)	0
Mucinous	9 (1.6)	6 (1.2)
Metaplastic	4 (0.7)	1 (0.2)
Apocrine	4 (0.7)	4 (0.8)
Papillary (invasive)	1 (0.2)	
GD2 percentage (%; median, IQR)	1 (0–5)	1 (0–3)
GD2 intensity		
0	283 (49.8)	152 (30.2)
1	63 (11.1)	113 (22.5)
2	105 (18.5)	82 (16.3)
3	117 (20.6)	156 (31.0)

BMI, body mass index; IDC, invasive ductal carcinoma; IF, immunofluorescence; IHC, immunohistochemistry; ILC, invasive lobular carcinoma; IQR, interquartile range; NST, no special subtype; TNBC, triple-negative breast cancer; SD, standard deviation.

**Table 2 cancers-13-05577-t002:** Proportions of samples among the breast cancer subtypes that expressed GD2 at the individual intensities 1, 2, or 3 on immunohistochemistry or immunofluorescence.

Intensity	Breast Cancer Subtypes				*p* *
	TNBC	Luminal A	Luminal B	HER2	
GD2 immunohistochemistry					
0	62 (81.6)	92 (40.4)	93 (47.0)	36 (54.5)	<0.0001
1	5 (6.6)	33 (14.5)	20 (10.1)	5 (7.6)	
2	4 (5.3)	44 (19.3)	46 (23.2)	11 (16.7)	
3	5 (6.6)	59 (25.9)	39 (19.7)	14 (21.2)	
GD2 immunofluorescence					
0	38 (55.1)	43 (21.9)	46 (25.3)	25 (44.6)	<0.0001
1	14 (20.3)	43 (21.9)	44 (24.2)	12 (21.4)	
2	8 (11.6)	36 (18.4)	29 (15.9)	9 (16.1)	
3	9 (13.0)	74 (37.8)	63 (34.6)	10 (17.9)	

Values are *n* (%). * Based on the Kruskal–Wallis test. TNBC, triple-negative breast cancer.

**Table 3 cancers-13-05577-t003:** Unadjusted and adjusted hazard ratios for GD2 immunohistochemistry and GD2 immunofluorescence values for the outcomes of disease-free survival and overall survival.

Outcome	Biomarker	Unadjusted HR (95% CI)	*p* Value	Adjusted HR * (95% CI)	*p* Value
GD2 immunohistochemistry					
DFS	GD2 binary (0 vs. >0)	0.74 (0.54, 1.02)	0.07	0.79 (0.56, 1.10)	0.16
	GD2 percentage	1.00 (0.99, 1.01)	0.94	1.00 (0.99, 1.01)	0.76
OS	GD2 binary (0 vs. >0)	0.77 (0.54, 1.10)	0.15	0.84 (0.57, 1.23)	0.36
	GD2 percentage	1.00 (0.99, 1.01)	0.67	1.00 (0.99, 1.01)	0.93
GD2 immunofluorescence					
DFS	GD2 binary (0 vs. >0)	1.18 (0.80, 1.73)	0.40	1.12 (0.75, 1.67)	0.59
	GD2 percentage	1.00 (0.98, 1.01)	0.65	0.99 (0.98, 1.01)	0.27
OS	GD2 binary (0 vs. >0)	1.05 (0.69, 1.61)	0.82	0.99 (0.63, 1.55)	0.96
	GD2 percentage	1.00 (0.98, 1.02)	0.94	0.99 (0.98, 1.01)	0.54

CI, confidence interval; DFS, disease-free survival; HR, hazard ratio; OS, overall survival. * HRs are adjusted for age at diagnosis, body mass index, tumor stage, lymph-node status, and subtypes.

## Data Availability

Data can be obtained on request. Data are not stored on publicly available servers.

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
