# Peer review of "Variable Expression of the Disialoganglioside GD2 in Breast Cancer Molecular Subtypes"

_cancers, 2021, doi:10.3390/cancers13215577_

Round 1
Reviewer 1 Report
I thank the authors for their work. The message of the manuscript seems to be that there is expression of GD2 in a proportion of breast cancers, which is in line with previous work by others. This is a modest claim, however I still have some concerns:
1. Patient selection: in Figure 1, the numbers 51 and 50 do not add up to 103.
2. Patient selection: why were 221 (IHC) and 286 (IF) patients not included for the GD2 expression analysis? My concerns here include selection bias or confounding factors.
3. The study adds to existing publications by analyzing more patient samples with two assays (IHC and IF), however only 1 antibody was used and in my opinion the study lacks validation. The specificity and sensitivity of the antibody for detecting GD2 in FFPE tissue could be validated better if well-characterized biological positive and negative controls can be found or generated (for example, genetically engineered GD2+ and GD2- human cancer cells xenografted in immunodeficient mice). Western blotting may also add to the validation of the results.
4. Lines 43 and 49: "was associated with" and "not significantly associated"; the wording of the sentences could be interpreted as contradictory statements, this can easily be rephrased to avoid confusion.
5. Table 1 is titled "Baseline characteristics" but also includes results. The table could be titled accordingly, or could be split.
6. The authors call the cytoplasmic granular signal "Golgi-like". The subcellular localization of the Golgi apparatus is paranuclear, not perinuclear or evenly distributed in the cytoplasm. I think to support this interpretation, more evidence would be needed. Perhaps the authors can add a paragraph or a few sentences about known possible subcellular localizations of GD2 in scientific literature.
7. line 308: "in only 1% and 1-2% of cases" of what cases? This sentence is unclear.
Author Response
“I thank the authors for their work. The message of the manuscript seems to be that there is expression of GD2 in a proportion of breast cancers, which is in line with previous work by others. This is a modest claim, however I still have some concerns:”
We thank the reviewer for taking his/her time to read our manuscript thoroughly and sharing his/her expertise with us. Furthermore, we appreciate the comprehensive review and constructive suggestions that have led to an improvement of the value and statements of the manuscript.
According to the recommendations, we have modified the manuscript (changes are tracked).
“1. Patient selection: in Figure 1, the numbers 51 and 50 do not add up to 103.”
We apologize for the mistake. Besides 51 and 50 patients, respectively, ineligible due to bilateral breast cancer or metastasis at initial diagnosis, 2 patients were excluded due to insufficient survival time (DFS/OS less than 1 day). This exclusion criterion was added to Figure 1 (page 4).
“2. Patient selection: why were 221 (IHC) and 286 (IF) patients not included for the GD2 expression analysis? My concerns here include selection bias or confounding factors.”
We endorse the criticism of the reviewer. Drop-out of 221 (IHC) and 286 (IF) cases can be explained by the use of tissue micro arrays. If there was no/inadequate invasive tumor tissue recognizable, if tumor core had been washed off during staining procedure or if there were staining artifacts (e.g. autofluorescence for IF) hampering the IHC/IF assessment, cases were excluded from staining evaluation. Hence, exclusion was rather caused randomly than due to any systemic selection bias. Nevertheless, we understand the concern and have mentioned this limitation within the discussion (page 13, line 405-407).
“3. The study adds to existing publications by analyzing more patient samples with two assays (IHC and IF), however only 1 antibody was used and in my opinion the study lacks validation. The specificity and sensitivity of the antibody for detecting GD2 in FFPE tissue could be validated better if well-characterized biological positive and negative controls can be found or generated (for example, genetically engineered GD2+ and GD2- human cancer cells xenografted in immunodeficient mice). Western blotting may also add to the validation of the results.”
We appreciate the appropriate comment, we understand the reviewer’s concerns. Indeed, we used a monoclonal antibody (clone 14.G2a) that has been developed for flow cytometry, IF, and IHC (on frozen tissue). We used this antibody due to several reasons. This antibody is used for therapeutic purposes. Furthermore, it is and will be used for flow cytometry in our university hospital (a clinical study that tests GD2 CAR T cell therapy in breast cancer patients is planned). Hence, we did not want to switch antibodies. Furthermore, IHC staining is much more suitable for daily routine diagnostics and available in (small) pathological laboratories that handle mostly with FFPE tissue. Hence, we wanted to investigate whether this antibody can also be used using IHC in FFPE tissue. To analyze this and to compare both methods, we assessed both IF and IHC. As shown in our study, both methods are discrepantly reporting GD2 expression. Hence, further studies are needed to find the best way for GD2 assessment in breast cancer. This issue is already mentioned in the limitations of our studies (page 13, line 410-412). For IHC, FFPE tissue from a neuroblastoma and from an invasive GD2-positive breast cancer sample were used as positive controls. As a negative control, buffer was applied in-stead of the antibody. For IF, sections of GD2-positive Ewing sarcomas and breast cancers were used as positive controls (see Methods, page 5, lines 145-146 and 165-166, respectively). The suggestion of the reviewer to use genetically engineered GD2+ and GD2- human cancer cells xenografted in immunodeficient mice or Western blotting is reasonable and we encourage other working groups to validate the antibody using these methods. However, due to limitations in regards of animal research in our working group and the limited revision time of 10 days, further in vivo/in vitro experiments are not feasible for this project.
“4. Lines 43 and 49: "was associated with" and "not significantly associated"; the wording of the sentences could be interpreted as contradictory statements, this can easily be rephrased to avoid confusion.”
We have changed the wording (page 1, line 43).
"5. Table 1 is titled "Baseline characteristics" but also includes results. The table could be titled accordingly, or could be split.”
Thank you, we have adjusted the heading accordingly (page 7, line 224).
“6. The authors call the cytoplasmic granular signal "Golgi-like". The subcellular localization of the Golgi apparatus is paranuclear, not perinuclear or evenly distributed in the cytoplasm. I think to support this interpretation, more evidence would be needed. Perhaps the authors can add a paragraph or a few sentences about known possible subcellular localizations of GD2 in scientific literature.”
This aspect is correct. First of all, we agree with the comment that perinuclear is the incorrect phrasing. We have modified the sentence (paranuclear instead of perinuclear). Unfortunately, there is only limited literature on GD2 in breast cancer and only few working groups describe the expression pattern in detail: Orsi et al. recognized “originally […] a cytoplasmatic along with a membrane staining by IHC”. However, they did not specify the cytoplasmic pattern in more detail. In the paper of Schulz et al. from 1984 (PMID: 6498849), Figure 1 shows some similarities in regards of the GD2 staining pattern in neuroblastoma cells when compared to our findings. However, the authors did not mention the subcellular location within their explanations and the figure is black-and-white.
Furthermore, we agree, that the exact distribution of GD2 expression within breast cancer cells should be investigated in further detail in the future. We added a comment to the study limitations (page 13, line 420-425).
“7. line 308: "in only 1% and 1-2% of cases" of what cases? This sentence is unclear.”
We apologize for this clumsy wording. We have changed the phrasing to make the intention clear (page 11, line 314-315). It has to be “Within these positive cases, 17/31 (54.8%) and 24/31 of the samples (77.4%) showed expression of GD2 in only 1% and 1–2% of all assessable epithelial cells, respectively.”
Reviewer 2 Report
This is an interesting and well-written manuscript on the expression patterns of the disialoganglioside GD2 across the molecular and morphologic subtypes of breast cancer.
I have no major objections but the following issues must be addressed:
- The tumor grade is mentioned in Materials and Methods but not in the results; it is a critical variable that should be used in the analysis.
- Some morphologic subtypes of breast cancer cannot be merged but analyzed separately (micropapillary and papillary). They represent strikingly different subtypes of breast cancer!
- You have a surprisingly low percentage of HER2+ breast cancers, could you please clarify why?
- Figure 2 is nice but the readers (particularly non-pathologists) would benefit from low magnification IHC images.
- How did you define ER, PR and HER2 positivity? Please add in the Materials and Methods.
Author Response
“This is an interesting and well-written manuscript on the expression patterns of the disialoganglioside GD2 across the molecular and morphologic subtypes of breast cancer. I have no major objections but the following issues must be addressed:”
We thank the reviewer very much for his/her time expertise and time. We appreciate the CONS the reviewer has listed and hope that we have modified the manuscript to his/her satisfaction.
- “The tumor grade is mentioned in Materials and Methods but not in the results; it is a critical variable that should be used in the analysis.”
We understand the point of criticism. In regards of molecular subtyping, grading was used to further divide luminal tumors into Luminal A (grade 1-2) versus Luminal B (grade 3) (please see Supplementary Materials, chapter “Immunohistochemical staining of ER, PR, HER2, and Ki-67”). However, we agree with the concern and have added the distribution of tumor grading of the study cohort within Table 1 (page 7).
- Some morphologic subtypes of breast cancer cannot be merged but analyzed separately (micropapillary and papillary). They represent strikingly different subtypes of breast cancer!
We thank the reviewer for this adequate comment. We totally agree, micropapillary and invasive papillary carcinoma are distinct histological subtypes. As recommended, we have changed Table 1 (page 7).
- You have a surprisingly low percentage of HER2+ breast cancers, could you please clarify why?
Expected HER2+ BC rates are 12-15 % (1. Slamon DJ, Clark GM, Wong SG, Levin WJ, Ullrich A, McGuire WL. Human breast cancer: correlation of relapse and survival with amplification of the HER-2/neu oncogene. Science. 1987;235(4785):177-82. Epub 1987/01/09. PubMed PMID: 3798106. 2. Slamon DJ, Godolphin W, Jones LA, Holt JA, Wong SG, Keith DE, et al. Studies of the HER-2/neu proto-oncogene in human breast and ovarian cancer. Science. 1989;244(4905):707-12. Epub 1989/05/12. PubMed PMID: 2470152. 3. Wolff AC, Hammond ME, Schwartz JN, Hagerty KL, Allred DC, Cote RJ, et al. American Society of Clinical Oncology/College of American Pathologists guideline recommendations for human epidermal growth factor receptor 2 testing in breast cancer. Archives of pathology & laboratory medicine. 2007;131(1):18-43. Epub 2007/01/01. doi: 10.1043/1543-2165(2007)131[18:Asocco]2.0.Co;2. PubMed PMID: 19548375.). In our study cohort, up to 11.6 % were of the HER2 subtype. Hence, the HER2+ subgroup is nearly the real-world range of HER2+ cases.
- Figure 2 is nice but the readers (particularly non-pathologists) would benefit from low magnification IHC images.
Thank you for the comment. Since we were not able to find distinct figures of GD2 staining in high magnification (in particular for GD2 IHC), when reviewing the literature, we decided to show high magnification images. Hence, the reader interested into detailed staining pattern is able to see the details. We hope that the reviewer accepts our approach.
- How did you define ER, PR and HER2 positivity? Please add in the Materials and Methods.
Details on ER/PR/HER2 assessment/interpretation are described in the Supplementary Materials, chapter “Immunohistochemical staining of ER, PR, HER2, and Ki-67”. Briefly, positive staining for ER and PR was time-dependently defined as ≥ 10% and ≥ 1%, respectively. HER2 IHC score was documented in the pathology reports as 0, 1+, 2+, or 3+ in accordance with the published guidelines. Tumors with a score of 0 or 1+ were considered as HER2-negative and those with a score of 3+ were defined as HER2-positive. Breast cancer samples with a 2+ staining were analyzed for gene copy numbers of HER2 using chromogenic in situ hybridization (CISH). A case was regarded as HER2-amplified if the HER2/CEN17 ratio was ≥ 2.2 (study cohort: 1997-2007; ratio according to guidelines of 2007). Before 2002, patients were retrospectively identified as being HER2-positive or -negative.
Reviewer 3 Report
Erber et following manuscript "Variable expression of the disialoganglioside GD2 in breast cancer molecular subtypes" overall findings are not very significant for TNBC but have higher expression in Other luminal breast cancers. Luminal B type patients are PR+, ER+ and HER2+ in that case how the author differentiating false positive for HER2+. Did the author also evaluate ER+ and PR+ categories? Overall scientific findings are not promising.
Author Response
“Erber et following manuscript "Variable expression of the disialoganglioside GD2 in breast cancer molecular subtypes" overall findings are not very significant for TNBC but have higher expression in Other luminal breast cancers. Luminal B type patients are PR+, ER+ and HER2+ in that case how the author differentiating false positive for HER2+. Did the author also evaluate ER+ and PR+ categories? Overall scientific findings are not promising.”
We thank the reviewer for reading our manuscript thoroughly and sharing his/her concerns with us. In regards of molecular subtyping, we defined subtypes as follows (please see Supplementary Materials, chapter “Immunohistochemical staining of ER, PR, HER2, and Ki-67”): The definitions of the subtypes have been reported previously (Wunderle et al. Association between breast cancer risk factors and molecular type in postmenopausal patients with hormone receptor-positive early breast cancer. Breast cancer research and treatment 2019, 174, 453-461). If the tumor had a HER2 IHC score of 3+ or showed amplification of the HER2 gene, HER2 status was considered positive (HER2-positive/HER2+ breast cancer) (Bethune et al. Impact of the 2013 American Society of Clinical Oncology/College of American Pathologists guideline recommendations for human epidermal growth factor receptor 2 (HER2) testing of invasive breast carcinoma: a focus on tumours assessed as 'equivocal' for HER2 gene amplification by fluorescence in-situ hybridization. Histopathology 2015, 67, 880-887). Patients with negative ER, PR, and HER2 status were defined as having triple-negative breast cancer (TNBC). HER2-negative breast cancers with expression of either ER or PR were further separated into luminal A (-like) tumors (grading of 1 or 2) and luminal B (-like) tumors (grading of 3) (von Minckwitz et al. Definition and Impact of Pathologic Complete Response on Prognosis After Neoadjuvant Chemotherapy in Various Intrinsic Breast Cancer Subtypes. Journal of Clinical Oncology 2012, 30, 1796-1804). Due to limited number of cases within the different intrinsic subgroups mentioned above, we decided not to further divide HER2+ cases into HER2+/luminal versus HER2+/non-luminal. Furthermore, we also did not divide luminal cases by ER and PR expression due to the same reason.
Round 2
Reviewer 1 Report
I thank the authors for their thorough revision of the manuscript. I still have a remaining concern regarding the lack of validation of the specificity (and sensitivity) of the antibody for IHC. The discrepancies between IHC and IF results underline this. The authors were given insufficient time (10 days) by the journal editors to resolve this and I think this is undermining to scientific quality. It is unclear to me whether the authors can find access to tumor tissue that could be used for protein analysis to validate IHC results via an additional method to quantify GD2 protein expression. In my view the entire application and presentation of the IHC as a potential clinical assay hinges upon such validation.
Round 3
Reviewer 1 Report
The authors have addressed the review comments in a fair manner, and I thank them for their efforts and good work. One minor caveat: the authors state that the GD2 antigen is tumor-specific, while from the data and the information shared in the introduction of the manuscript, it follows that GD2 can be expressed in normal tissues. However, this is just a consideration I would like to share with the authors. I recommend publication of the manuscript.
This manuscript is a resubmission of an earlier submission. The following is a list of the peer review reports and author responses from that submission.
Round 1
Reviewer 1 Report
In this study, the authors investigated GD2 expression on breast cancer tissues using two types of methods; immunohistochemistry (IHC) and immunofluorescence (IF). And they also examined its impact on survival among breast cancer cohort (n=894). Although they tackled the unknown of this area to surpass the limitation and the insufficiency in the previous studies like a smaller size of samples, a study only focusing on a prognostic role of a single molecule would be considered to have less scientific significance.
Major points;
-Eventually, the independent prognostic value of GD2 has not been proven in breast cancer cohort. Furthermore, the survival difference between higher and lower groups of GD2 might reflect the survival difference between subtypes, especially luminal BC and TNBC. At this point, the independent prognostic value is relevant.
-The basic experiments including a proliferation one using GD2 overexpressed cultured cells and anti-GD2 antibodies or GD2 CAR T cells could support your result and hypothesis.
Minor points;
-The labels of Figure (1, 2, 3, ) seem not to match to the explanation, I could not find the box plots the authors mentioned in the manuscript.
-TBNC should be TNBC (L. 357).
Reviewer 2 Report
This paper analyzes the role of disialoganglioside GD2 in a large series of breast cancer. Using two different staining methods (immunohistochemistry, IHC, and immunofluorescence, IF), the authors found a high expression of GD2 in breast cancer (over 50% of cases). Interestingly, GD2 was differentialy expressed among different breast cancer subtypes, with highest proportions of GD2-positive tumors in luminal breast tumors. No significant association was found between GD2 expression and outcome of patientes (disease-free or overall survival). The authors conclude that patients with GD2-positive advanced breast cancer could benefit from GD2-targeting immunotherapies.
My main concern relates to the evaluation of GD2. IHC was performed manually using a monoclonal antibody (clone 14.G2a) that has been developed for flow cytometry, IF, and IHC (on frozen tissue). How this may influence the results obtained in the immunohistochemical study remains to be determined. In fact, in the previous studies indicated in the references (see refs. 16, 19, 41, and 42), the antibody was used either in flow cytometry studies from cell suspensions or in immunohistochemical studies but in frozen tissue samples, not on formalin-fixed, paraffin-embedded tissue samples.
The differences in the results obtained in the two techniques used to evaluate the expression of GD2 is very striking. In fact, the correlation between both techniques is in the low range of what is considered a good correlation. When the overall results are analyzed, 50.2% and 69.8% of breast cancer samples showed GD2 expression at different intensity levels by IHC and IF, respectively. The distribution of GD2 intensities also differed significantly among breast cancer subtypes with both staining methods (18.5%-IHC vs 44.9%-IF in TNBC, 59.7%-IHC vs 78.1%-IF in LA, 53%-IHC vs 74.7%-IF in LB, and 45.5%-IHC vs 55.4%-IF in HER2 tumors). Possibly, these striking differences reflect the inconsistency of GD2 assessment in breast cancer samples using the methods applied in this study and make the results obtained not very reproducible.
Another difficulty has to do with the assessment of immunohistochemical staining. As the authors indicate, GD2 IHC shows a cytoplasmic, mostly perinuclear granular pattern (“Golgi-like” pattern), and a distinct membranous staining pattern clearly different from the cytoplasmic expression was not distinguisable on IHC. How can the authors be sure then that in all the cases they are evaluating there is actually membrane expression and not just cytoplasm? It is true that the membrane expression of GD2 was confirmed by flow cytometry in 4 samples that previously tested positive for GD2 expression on IHC/IF but I am not sure that this is enough to guarantee that whenever granular staining is observed in the cytoplasm of the tumor cells this necessarily means that there is also membrane staining.
Regarding the assessment of GD2 expression, why not combine intensity and percentage of GD2-positive tumor cells in a score similar to the Allred-score for evaluation of hormone-receptors in breast cancer?
In line with all of the above, other important limitations of the study, in addition to those indicated by the authors, are the absence of external validation of the GD2 IHC staining and scoring system. In my opinion, the conclusions stated in the discussion regarding this study are formulated too strong and, therefore, the authors should weaken their statements.
Additional comments
I understand that the HER2 subtype in this study includes only pure HER2 cases; on the other hand, luminal B phenotype can be subdivided into luminal B-HER2 positive and luminal B-HER2 negative. Authors should take this into account and redefine breast cancer subtypes accordingly.
Since this study does not find any association with the prognosis of the patients, the title of the article is somewhat confusing. I suggest removing the part that refers to the prognosis.
Abstract, line 41. GD2 expression was not assessed in a cohort of 894 breast cancer patients; this figure (n=894) corresponds to the cohort from which the study is proposed, but the cohort in which GD2 expression is analyzed is actually 568 patients (IHC) and 503 patients (IF). Therefore, this figure should be corrected to reflect what was actually done in the study. The same applies to the first sentence of the Discussion (line 312).
There are to minor typos across the text: line 217 (789 instead of 798) and line 255 (55.4% instead of 54.4%).